# Aminoglycosides for the Treatment of Severe Infection Due to Resistant Gram-Negative Pathogens

**DOI:** 10.3390/antibiotics12050860

**Published:** 2023-05-06

**Authors:** Michaël Thy, Jean-François Timsit, Etienne de Montmollin

**Affiliations:** 1Assistance Publique Hôpitaux de Paris (AP-HP), Service de Médecine Intensive et Réanimation Infectieuse, Hôpital Bichat Claude-Bernard, Université Paris Cité, 46 Rue Henri Huchard, 75018 Paris, France; michael.thy@aphp.fr (M.T.); etienne.demontmollin@aphp.fr (E.d.M.); 2Equipe d’accueil (EA) 7323, Department of Pharmacology and Therapeutic Evaluation in Children and Pregnant Women, Université Paris Cité, 75018 Paris, France; 3Unité mixte de Recherche (UMR) 1137, Infection, Antimicrobials, Modelization, Epidemiology (IAME), Institut National de la Recherche Médicale (INSERM), Université Paris Cité, 75018 Paris, France

**Keywords:** aminoglycosides, pharmacokinetics, multidrug-resistant, gram-negative infections, toxicity, nebulization

## Abstract

Aminoglycosides are a family of rapidly bactericidal antibiotics that often remain active against resistant Gram-negative bacterial infections. Over the past decade, their use in critically ill patients has been refined; however, due to their renal and cochleovestibular toxicity, their indications in the treatment of sepsis and septic shock have been gradually reduced. This article reviews the spectrum of activity, mode of action, and methods for optimizing the efficacy of aminoglycosides. We discuss the current indications for aminoglycosides, with an emphasis on multidrug-resistant Gram-negative bacteria, such as extended-spectrum β-lactamase-producing *Enterobacterales*, carbapenemase-producing Enterobacterales, multidrug-resistant *Pseudomonas aeruginosa*, and carbapenem-resistant *Acinetobacter baumannii*. Additionally, we review the evidence for the use of nebulized aminoglycosides.

## 1. Introduction

Aminoglycosides (AGs) represent a class of antibiotics that were first discovered in the mid-1940s [1] and continue to be used extensively for the treatment of severe infections. Streptomycin, the inaugural AG, was initially developed as an anti-tuberculosis agent [1]. Over time, several other AGs were identified, encompassing three molecules that remain commonly employed in contemporary medicine: gentamicin (1963), tobramycin (1967), and amikacin (1972) [2,3,4]. Plazomicin, a next-generation aminoglycoside (2010), has an interesting bactericidal activity against most aminoglycoside-resistant strains such as extended-spectrum β-lactamase (ESBL) and carbapenemase-producing *Enterobacterales* [5,6,7].

In the current medical landscape, AGs retain their status as an important component in the management of sepsis, owing to their rapid and potent bactericidal activity [8]. Furthermore, they demonstrate preserved minimum inhibitory concentrations (MICs) in difficult-to-treat (DTR) Gram-negative bacteria (GNB) such as *Pseudomonas aeruginosa* and *Acinetobacter baumannii*, as well as in pathogens with acquired resistance, including ESBL *Enterobacterales* [9]. Indeed, a recent international survey indicated that short courses of AGs continue to be widely administered to critically ill patients with septic shock, despite wide variability in AG use [10]. Due to their narrow therapeutic index, which entails risks of nephrotoxicity and ototoxicity, it is essential that AGs be prescribed within a strict framework of indications and administration schemes accompanied by careful monitoring.

In this review, we will discuss the advantages and pitfalls of the most commonly used AGs in the treatment of severely resistant Gram-negative organisms, namely amikacin, gentamicin, and tobramycin, as well as plazomicin, a promising next-generation aminoglycoside.

## 2. Mechanism of Action and Spectrum of Activity

The potent bactericidal activity exhibited by AGs can be attributed to their specific binding to the 30S ribosomal subunit, which effectively disrupts bacterial intracellular protein synthesis [11]. Gentamicin and tobramycin exhibit comparable antimicrobial properties. In contrast, amikacin displays higher MIC, which is offset by a reduced level of nephrotoxicity, thus allowing for higher serum concentrations.

Although gentamicin, tobramycin, and amikacin present a mostly similar microbiological spectrum of activity, some distinctions can be observed. For Enterobacterales, the three agents display comparable efficacy, with the exception of amikacin, the only AG active on *Providencia* spp., and gentamicin, to be preferred on *Serratia marcescens* [4]. ESBL-producing strains remain susceptible to gentamicin and amikacin in approximately 50% and 70% of cases, respectively [9,12,13,14]. Consequently, amikacin is the optimal candidate for the empirical treatment of nosocomial infections or in situations where the prevalence of ESBL-producing strains is high. Regarding *P. aeruginosa*, tobramycin exhibits the most potent bactericidal activity among AGs and the lowest resistance rates [3]. For *A. baumannii*, both amikacin and tobramycin are the most frequently effective molecules [15]. It is important to note that AGs are inherently inactive against *Stenotrophomonas maltophilia* and anaerobes [16,17].

Plazomicin has the potential to fill a unique role in antimicrobial therapy due to the limited therapeutic options available for treating multidrug-resistant (MDR) pathogens. While the majority of data focus on complicated urinary tract infections [6,18], a multicenter, randomized, open-label trial has demonstrated that a definitive combination-therapy regimen with plazomicin, as compared to colistin, is effective in treating serious infections caused by carbapenem-resistant Enterobacterales. These infections include bloodstream infections and hospital-acquired or ventilator-associated bacterial pneumonia [19]. These promising results have led to approval by the European Medicines Agency and the Food and Drug Administration, but the drug has not yet been introduced to the market.

## 3. Main Side Effects and Resistance to Aminoglycosides

The main reported toxicities of AGs are nephrotoxicity, ototoxicity, and neuromuscular blockade. Nephrotoxicity occurs in 3–11% of patients, while vestibular toxicity occurs in 10% and cochlear toxicity in 26% (Table 1). The risks of renal and cochleovestibular toxicities increase with treatments exceeding 5 to 7 days, even in healthy individuals, and are higher in patients with chronic kidney disease. Renal toxicity is not related to peak plasma concentration, and no data correlate peak levels with auditory and vestibular toxicity, even when administered as a once-daily dosing (ODD). In most situations, however, AG therapy may be discontinued after 48 to 72 h (the approximate time needed to obtain the results of the antibiogram). 

Clinicians must be aware of ESBL risk factors and coresistance to select appropriate antibiotics that may avoid the unnecessary use of aminoglycosides [20]. The two fundamental mechanisms of resistance to aminoglycosides in *Enterobacterales* are enzymatic modification or modification of their target [21,22]. Aminoglycoside-modifying enzymes are grouped into three large families: nucleotidyl (adenyl)-transferases, phosphotransferases, and acetyltransferases (AACs), each with numerous variants. The aac(6′) gene encodes an acetyltransferase enzyme that modifies the aminoglycoside molecule, preventing the drug from binding to the bacterial ribosome and reducing its antibacterial activity. The aac(6′) gene is frequently found in Gram-negative bacteria, particularly *Escherichia coli* and *Klebsiella pneumoniae*. Resistance can also occur through impermeability or active elimination by efflux pumps (AcrD) [23]. Each specific enzyme affects certain aminoglycosides but not others. The new compound plazomicin evades nearly all modifying enzymes of this kind [24]. Moreover, a single microorganism can produce multiple enzymes from the same or different families. On the other hand, a mutation of the two main methylase families inactivates all aminoglycosides of clinical interest (even plazomicin), resulting in high-level resistance. The group of Gram-negative and Gram-positive bacteria caused by nosocomial infections called ESKAPE (*E. faecium*, *Staphylococcus aureus* (*S. aureus*), *Klebsiella pneumoniae* (*K. pneumoniae*), *A. baumannii*, *P. aeruginosa*, and *Enterobacter*) are highly resistant to antibiotics including AG [25].

The epidemiological cut-off values for gentamicin, amikacin, and tobramycin vary depending on the bacterial species. For gentamicin cut-off values are 2 mg/L for *Enterobacterales* and *A. baumannii*, and 8 mg/L for *P. aeruginosa*. For amikacin, the cut-off values are 8 mg/L for *Enterobacterales* and *A. baumannii*, and 16 mg/L for *P. aeruginosa*. For tobramycin, the cut-off values range between 2 and 4 mg/L for *Enterobacterales* and *A. baumannii*, and are 2 mg/L for *P. aeruginosa* [26]. Resistance to aminoglycosides greatly varies among bacterial species and geographical regions. In the report on AMR published jointly by the European Centre for Disease Prevention and Control (ECDC) and the World Health Organization (WHO) Regional Office for Europe, resistance to AGs (gentamicin/netilmicin/tobramycin) was greatly variable among countries. Resistance rate ranged from 4.1 to 27.0% for *E. coli*, from 0 to 69.1% for *K. pneumoniae*, from 0 to 41.7% for *P. aeruginosa*, and from 2.1 to 98.8% for *Acinetobacter* spp. [12]. In a retrospective cohort of Gram-negative bacteremia in 173 US hospitals, drug-resistant (DTR) isolates were non-susceptible to gentamicin (74%), tobramycin (71%), and amikacin (54%), all higher than among the CDC-defined phenotypes [27]. Concerning the susceptibility results for these three AGs (n = 28,259), 1% overall versus 33% of DTR isolates displayed class resistance. Only 14% of carbapenem-resistant, 4% of extended-spectrum cephalosporin-resistant, and 0.5% of fluoroquinolone-resistant episodes were resistant to all AGs.

In the Asia−Pacific region, for carbapenem-resistant, multi-drug-resistant, and difficult-to-treat *A. baumannii*, the rate of resistance to amikacin was 87%, which remained the most effective agent in vitro [28]. In Europe, amikacin also demonstrated the highest activity in most regions, with susceptibility rates between 11% and 38% [29]. Among MDR *A. baumannii*, susceptibility varied from 19% in Latin America and Europe to 62% in the United States [29].

## 4. Pharmacokinetic/Pharmacodynamic (PK/PD) Properties of Aminoglycosides and Optimization for the Critically Ill Patient

### 4.1. Pharmacokinetic Properties

AGs consist of amino sugars and aminocyclic alcohols [30], demonstrating high water solubility [31]. Pharmacokinetic properties are similar across all AGs. Consistent with their hydrophilic nature, AGs exhibit a low volume of distribution (Vd) (0.3 to 0.4 L/kg), exclusive renal clearance without metabolization, and limited intracellular penetration [32]. Consequently, AGs have mediocre diffusion in tissues and fluids such as the central nervous system, bronchial secretions, or aqueous humor. For instance, gentamicin has an epithelial lining fluid penetration ratio of 0.32, meaning that only a third of the plasma concentration can be expected in the alveoli [33]. In contrast, the epithelial lining fluid penetration ratios of cefepime and levofloxacin are 1 and 1.3, respectively [34,35]. Protein binding is low at approximately 20%, and the elimination half-life is 2 to 3 h in patients with normal renal function.

### 4.2. Pharmacodynamic Properties

The bactericidal activity of AG is concentration-dependent, and optimal therapeutic effect is achieved when the ratio of peak concentration (C_max_) over the MIC is between 8 and 10. In other words, peak plasma concentrations should be at least 8 to 10 times the MIC of the suspected bacteria [36,37]. This bactericidal activity is associated with a prolonged postantibiotic effect and less adaptive resistance to the first dose [38,39].

### 4.3. Optimization for the Critically Ill Patient

As AG efficacy is concentration-dependent, they should be administered intravenously in a 30-min infusion to achieve maximal peak plasma concentrations. For the same reason, in treating GNB, AGs should only be administered as a once-daily dose [40]. This approach, resulting in a higher C_max_, offers several advantages: (1) improved likelihood of reaching pharmacodynamic targets, (2) better tissue diffusion due to a larger concentration gradient between plasma and target tissues, (3) decreased renal toxicity, and (4) an extended postantibiotic effect, lowering the risk of resistance emergence.

When administering probabilistic treatment to critically ill patients, clinicians should consider the worst-case scenario and target a C_max_ of at least 8–10 times the highest possible MIC. Therefore, the target C_max_ should be 32–40 mg/L for gentamicin/tobramycin and 64–80 mg/L for amikacin. Achieving these targets in critically ill septic patients is challenging, particularly as the volume of distribution of AGs increases in these conditions [32]. For instance, in critically ill patients receiving an amikacin dose of 25 mg/kg based on total body weight, one-third had a C_max_ below the target [41,42]. Factors associated with insufficient amikacin peak concentrations included a positive 24-h fluid balance and a body mass index (BMI) below 25 kg/m^2^ (using total body weight) [42]. When using adjusted body weight for dose calculation, a 30 mg/kg dosing regimen achieved pharmacodynamic success in 77% of cases [43]. Consequently, the dose of amikacin should be between 25 and 30 mg/kg, depending on the clinical situation and the choice of weight used for dose calculations. Despite limited data for gentamicin/tobramycin, a 5 to 8 mg/kg regimen can be proposed. For severely obese patients, utilizing adjusted body weight for dose calculation is suggested [44,45].

Decreased renal creatinine clearance itself does not significantly alter the Vd of AGs. Therefore, the weight-based dosage should not be adjusted according to creatinine clearance; only the time interval between injections will be increased, based on trough concentration (C_min_) monitoring. In patients undergoing renal replacement therapy (RRT), interval adaptation will depend on the RRT modality. For intermittent hemodialysis, it is preferable to administer the AG dose 2 to 4 h before dialysis to achieve a high peak concentration while minimizing side effects due to the rapid AG clearance during hemodialysis [46,47,48]. Dosing adjustment for patients receiving continuous RRT largely depends on the techniques employed (continuous filtration, dialysis, or both) and the effluent flow rate [49]. In critically ill patients undergoing continuous venovenous hemofiltration at 30 mL/kg/h, an extended interval with a high loading dose of amikacin of 25 mg/kg every 48 h has been recommended [50], along with therapeutic drug monitoring.

AG use warrants routine therapeutic drug monitoring due to its narrow therapeutic index and potential for nephrotoxicity and ototoxicity [51]. Peak plasma concentrations (C_max_) assess the effectiveness (fulfilling PK/PD objectives), and trough levels (C_min_) are predictive of renal toxicity. Peak plasma concentrations should be evaluated after the first injection in patients with severe presentations or in situations with probable pharmacokinetic parameters, such as septic shock, burns, febrile neutropenia, mechanically ventilated ICU patients, morbid obesity, multiple trauma, or cystic fibrosis. [32]. C_max_ measurement is conducted 30 min following the end of the 30-min infusion. Meanwhile, C_min_ should be assessed only if the treatment is expected to last over 5 days (measured after 48 h of treatment) or if creatinine clearance is compromised. Trough levels higher than 2.5 mg/L for amikacin or 0.5 mg/L for gentamicin/tobramycin require extending the interval between injections [52,53]. Figure 1 illustrates the main PK/PD challenges associated with AG treatment, and Table 2 presents an overview of general prescription guidelines and optimization strategies.

## 5. Aminoglycosides in the Empirical Treatment of Sepsis Due to Gram-Negative Bacteria

Combining AG with β-lactams in the empirical treatment of sepsis can broaden the spectrum of clinical treatment, expedite bacterial clearance, and reduce the emergence of antibiotic resistance due to their synergistic antibacterial effects [54,55]. The importance of selecting an optimal antibiotic therapy for sepsis is well recognized, and any delay between sepsis onset and the initiation of effective antimicrobial therapy is associated with a decreased probability of survival [56]. Despite these theoretical advantages, several randomized controlled trials (RCTs) and meta-analyses have failed to show the benefit of combination therapy over β-lactam monotherapy for GNB infections [57,58]. Furthermore, when considering AG specifically as an adjunctive antibiotic, a 2014 Cochrane meta-analysis showed no benefit of combination therapy over β-lactam monotherapy (including in the subgroup of *P. aeruginosa* infections) and was associated with an increased risk of acute kidney injury [59]. However, these data come from global, heterogeneous populations, and some subgroups may still benefit from combination therapy. For instance, studies focusing on patients with sepsis and septic shock with a predicted mortality of more than 25% showed the benefit of combination therapy [60,61]. Similarly, combination therapy improved appropriate antimicrobial therapy rates in patients with bloodstream infections due to carbapenem-resistant *Enterobacterales* (CRE) and increased survival in the most severe patients [62].

The Surviving Sepsis Campaign recommendations regarding the use of more than one Gram-negative agent for empirical therapy state that combination therapy should be restricted to patients at risk of infection with a multidrug-resistant (MDR) pathogen [63]. In this regard, the preferred additional antibiotic should be amikacin, as ESBL-producing GNB strains remain susceptible to this agent in 70% to 90% of cases [9,12,13]. Risk assessment for MDR GNB should include evidence of infection or colonization with antibiotic-resistant organisms in the past year, local prevalence of antibiotic-resistant organisms, hospital-acquired or healthcare-associated infection (as opposed to community-acquired infection), use of broad-spectrum antibiotics within the past 90 days, concurrent use of selective digestive decontamination, travel to a highly endemic country within the past 90 days (see https://resistancemap.cddep.org/ (accessed on the 5 May 2023) and hospitalization abroad within the previous 90 days [64,65,66]. Local information on the antimicrobial resistance profiles of the most common pathogens causing sepsis is also essential for the selection of the most appropriate empirical antibiotic therapy.

In neutropenic patients, the postantibiotic effect of AGs may also be helpful. In a large propensity-matched cohort study Albasanz-Puig et al. found that the initial combination therapy was associated with a lower 7-day case fatality rate (OR, 0.33; 95% CI, 0.13 to 0.82; *p* = 0.017) in neutropenic patients with Gram-negative bloodstream infection [67].

In summary, combining AG with β-lactams in sepsis treatment can offer potential benefits, but several RCTs and meta-analyses have not shown a clear advantage over β-lactam monotherapy. Some subgroups, such as patients with high predicted mortality rates or infections due to MDR pathogens, may still benefit from combination therapy. Current guidelines recommend combination therapy for patients at risk of infection with MDR pathogens, with amikacin as the preferred additional antibiotic.

## 6. Beyond Empirical Treatment, Is There a Role for Aminoglycosides in Definitive Therapy?

In almost all situations, AGs can be stopped after 48 to 72 h (which is about the time needed to obtain the results of the antibiogram). Theoretic advantages were described for synergism with β-lactams, especially for shorter in vitro time-to-kill but remain controversial with the latter data [68,69,70,71,72]. AGs may be continued for up to 5 days in the absence of microbiological documentation and specific settings, such as neutropenic patients in septic shock [61,67], but this practice is not evidence-based. The utility of AGs in definitive therapy must be addressed according to pathogens and resistance phenotypes: ESBL, CRE, MDR *P. aeruginosa*, and carbapenem-resistant *A. baumannii* (CRAB).

For ESBL *Enterobacterales* infections, there are no data to support the use of AGs beyond the empiric phase in severe infections. AGs have been suggested as an alternative option by the Infectious Diseases Society of America (IDSA) and the European Society of Clinical Microbiology and Infectious Diseases (ESCMID) for nonsevere infections [73,74]. For ESBL cystitis, for instance, a single intravenous AG dose is generally effective and exhibits minimal toxicity, although robust trial data remains limited [75,76,77]. For pyelonephritis or complicated urinary tract infections, AG may be considered an alternative if first-line options are unavailable or not tolerable [76,77]. For uncomplicated bloodstream infections with complete source control (e.g., urinary source or controlled sources, such as removal of an infected vascular catheter), IDSA suggests AG monotherapy when preferred treatment options are not accessible.

Regarding CRE infections, IDSA and ESCMID suggest that combination antibiotic therapy (i.e., a β-lactam agent with an AG) should not be routinely recommended for treating CRE infections if the pivotal antibiotic is effective in vitro (documented therapy) [73,74]. A meta-analysis revealed that, while monotherapy was often unsatisfactory, the efficacy of combination therapy remained uncertain [78]. AG are disadvantaged by their nephrotoxicity and thus considered second-line agents due to the availability of newer β-lactams and β-lactamase inhibitor combinations (BLBLI) such as ceftazidime-avibactam [79]. Although in vitro synergy was observed when polymyxins were combined with AG or carbapenems, clinical synergy remains unclear, primarily based on small observational studies [80,81,82]. To date, no RCT has examined whether combination therapy outperforms monotherapy for CRE infections. Observational studies indicate that combination therapy benefits are primarily observed in patients with serious underlying diseases or high pretreatment probability of death, such as septic shock [62,83,84,85,86]. However, determining the most effective regimen remains challenging [86]. Regarding CRE urinary infections, some studies found that AG demonstrated better clinical outcomes compared to polymyxins or tigecycline [87,88]. Similarly, AG showed a 78.9% clinical success rate compared to other antibiotics (37.0%, *p* = 0.007) in kidney transplant recipients with polymyxin-resistant CRE infections [87]. Overall, ESCMID guidelines consider the use of AGs as good clinical practice for patients with non-severe infections due to CRE, if active in vitro, on an individual basis, and according to the source of infection [74]. For patients with complicated UTI, AG, including plazomicin, was suggested over tigecycline [6,74]. For patients with severe infections caused by CRE strains susceptible in vitro only to polymyxins, AG, tigecycline, or Fosfomycin (or in the case of recent BLBLI unavailability), combination therapy was suggested without recommending specific associations [74].

Regarding *P. aeruginosa*, multidrug resistance is characterized by nonsusceptibility to at least one antibiotic in a minimum of three classes where susceptibility is generally expected: penicillins, cephalosporins, fluoroquinolones, AGs, and carbapenems [89]. For the treatment of severe infections caused by carbapenem-resistant *P. aeruginosa* with polymyxins, AG, or fosfomycin, ESCMID guidelines suggest treatment with two molecules active in vitro [74]. However, no recommendation was made for or against specific combinations. Although empirical combination antibiotic therapy (e.g., adding an AG or polymyxin to a β-lactam agent) to increase the likelihood of at least one active therapeutic for patients at risk for DTR *P. aeruginosa* infections is reasonable, data do not suggest that continued combination therapy, once the β-lactam agent has shown in vitro activity, provides additional benefits over β-lactam monotherapy [58,73]. 

In patients with severe and high-risk CRAB infections, ESCMID guidelines recommend combination therapy comprising two in vitro active antibiotics from available options (polymyxin, AG, tigecycline, or sulbactam combinations) [74]. Despite five RCTs and numerous meta-analyses addressing CRAB infection treatments, the optimal regimen remains undetermined [90,91,92,93,94,95,96]. Importantly, none of the RCTs showed a survival advantage with combination therapy that included AG [96].

In summary, AGs are not typically used for the definitive treatment of infections. AGs are suggested as an alternative option in ESBL *Enterobacterales* only for non-severe infections, while their role in CRE infections remains uncertain. For DTR *P. aeruginosa* and CRAB, available data suggest using two in vitro active molecules, though no specific combinations are recommended.

## 7. Benefits of Inhaled Aminoglycosides as Adjunctive Therapy

Ventilator-associated pneumonia (VAP) is associated with a high rate of treatment failure [97]. Potential causes of failure include high bacterial inoculum, poor lung diffusion of antibacterial agents, altered bacterial clearance, and impaired local immunity. The situation is even more complex when the MICs to the antimicrobial agent are high (close or beyond the resistance breakpoint) and increasing the antimicrobial dose is associated with increased toxicity. Nebulized AG may be an interesting option in these situations [98].

In a large systematic review including 1733 patients from 13 RCTs performed in mechanically ventilated patients, Qin et al. [99] found that nebulized amikacin was associated with a better microbiological eradication (RR = 1.51, 95% CI 1.35 to 1.69, *p* < 0.001) and better clinical response (RR = 1.23; 95% CI 1.13 to 1.34; *p* < 0.001) than controls, but without an impact on overall mortality (RR, 1.17 (0.91, 1.50), *p* = 0.77). Amikacin nebulization was associated with an increased risk of bronchospasm (RR, 2.55 (1.40, 4.66), *p* = 0.02), which may mitigate these positive results in severely hypoxemic patients. Importantly, a beneficial clinical response was not found in blinded studies or studies that used the recommended Mesh nebulizers. In the largest double-blind randomized trial included in the meta-analysis, Niederman et al. tested the effect of adjunctive Mesh nebulized amikacin in 807 patients with VAP [100]. The study failed to detect any difference in mortality or clinical cure. Mortality was similar even in cases with MDR GNB (amikacin 25/84 vs. placebo 19/79). With the inherent limitations of subgroup analyses, eradication of *P. aeruginosa* occurred in a greater proportion of patients receiving inhaled amikacin than placebo (73% vs. 50%; *p* = 0.003). Another single-center, double-blind study compared adjunctive therapy with 7 days of aerosolized amikacin (400 mg tid) versus a placebo administered via a jet nebulizer in MDR GNB (non-fermentative GNB n = 38, Enterobacterales n = 22). Adjunctive nebulized amikacin resulted in faster temperature control and improved oxygenation but had no effect on the delay of successful ventilator weaning or 28-day mortality. Bacterial eradication was more frequently achieved at the end of treatment with adjunctive nebulized amikacin (13/32 vs. 4/28, *p* = 0.024) without the emergence of amikacin resistance during the 28-day follow-up [101]. Finally, a randomized, double-blind, placebo-controlled, phase 2 study conducted in mechanically ventilated patients with pneumonia compared adjunctive inhaled amikacin and fosfomycin therapy delivered through a patented vibrating plate electronic nebulizer in addition to IV carbapenem with IV carbapenem alone [102]. Clinical improvement was not influenced by aerosolized antibiotics, but there was a trend toward more ventilator-free days and higher clinical cure rates on day 14 in the subset of 13 patients infected with highly resistant (amikacin MIC > 1024 mg/L) *Acinetobacter* spp. (66.7% vs. 25.0%, *p* = 0.16). 

Concerning AGs other than amikacin, a Bayesian network meta-analysis suggested that nebulized tobramycin may be more effective than amikacin in terms of clinical cure [98]. A prematurely stopped RCT on GNB VAP found a nonsignificant benefit of tobramycin (300 mg twice daily) over placebo for clinical cure (9/13 vs. 5/13, *p* = 0.24) with no benefit on mortality [103]. 

In summary, the modest benefits in bacterial clearance of highly resistant GNB observed with nebulized AGs do not necessarily lead to improved clinical outcomes and may be offset by poor tolerance of nebulization, particularly in patients with significant oxygenation impairment. Nebulized AG use should be restricted to ICUs experienced in antibiotic nebulization, employing checklists and specific monitoring to minimize misuse and adverse effects [104]. The risk/benefit ratio of nebulized AGs may be optimized when employed to target MDR GNB, preferably during the early treatment when inoculum levels and resistance emergence risks are high.

## 8. Conclusions

AGs represent an established class of antibiotics known for their powerful bactericidal properties and maintained effectiveness in treating MDR GNB. However, their narrow therapeutic index, primarily due to nephrotoxic side effects, has led to a gradual reduction in their range of applications. Presently, AGs are often administered in conjunction with another antimicrobial agent (usually a β-lactam) for the empirical treatment of patients with sepsis or septic shock and at high risk of MDR GNB infections. Generally, AG should not be utilized for the definitive treatment of severe infections, but they may still serve as a viable therapeutic alternative for addressing CRE, MDR *Pseudomonas*, or CRAB when other options are unavailable. If using AGs, clinicians must follow general rules based on their pharmacodynamic properties, which involve proper dosing, administration, and therapeutic drug monitoring, to ensure optimal effectiveness and minimal toxicity.

## Figures and Tables

**Figure 1 antibiotics-12-00860-f001:**
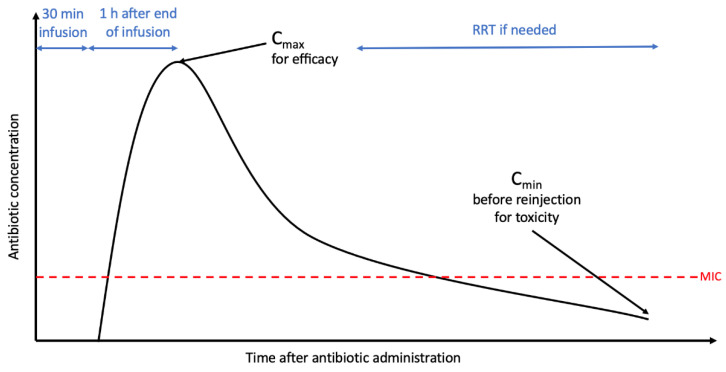
Main Pharmacokinetic/Pharmacodynamic issues for aminoglycosides. C_max_: peak concentration; C_min_: trough concentration; RRT: renal replacement therapy; double blue arrows corresponding to timing; dotted red line corresponding to MIC: minimal inhibitory concentration.

**Table 1 antibiotics-12-00860-t001:** Main side effects of aminoglycosides.

	Effect	Clinical Risk Factor	Treatment Risk Factor	Toxicity Prevention	Potential Treatment
Nephrotoxicity	Acute kidney injury with preserved diuresis, tubular necrosis	Chronic kidney disease, age, dehydration, hyperthermia	Cumulative dose, treatment duration > 5 days, previous AG treatment.	- Avoid cumulative risk factors - Avoid co-nephrotoxic treatments - Therapeutic drug monitoring (TDM)	- Dose adaptation through TDM - Stop AG when unnecessary
Cochleovestibular toxicity	- Vestibular: vertigo, ataxia, nystagmus - Cochlear: tinnitus, hearing loss	Previous hearing loss	Similar to nephrotoxicity		
Neuromuscular toxicity	Neuromuscular blockade	Myasthenia gravis Respiratory acidosis Immediate postoperative period	-		Anticholinesterase treatment

**Table 2 antibiotics-12-00860-t002:** Summary of the general rules for prescription and optimization of aminoglycosides.

Administration Modality [40]	Once-Daily Dose 30 min Intravenous Infusion
Dosage [32,41,42,43]	Gentamicin/tobramycin: 5–8 mg/kg Amikacin: 25–30 mg/kg Patients with BMI ≥ 30: the use of adjusted body weight is recommended
Impaired creatinine clearance and RRT [44,45,46,47,48,49,50]	No adaptation of based-weight dosage Increase in inter-dose interval Intermittent hemodialysis: prioritize administration 2–4 h before dialysis CVVH(D)F: suggested administration of 25 mg/kg every 48 h for amikacin.
Therapeutic drug monitoring [52,53]	Recommended for aminoglycosides C_max_/peak concentration (efficacy) Measure 30 min after the end of AG infusion Objective for gentamicin/tobramycin: 32–40 mg/L Objective for amikacin: 64–80 mg/L C_min_/trough concentration (toxicity) Measure before reinjection Objective for gentamicin/tobramycin: <0.5 mg/L Objective for amikacin: <2.5 mg/L

Abbreviation: RRT, renal replacement therapy; BMI, body mass index; CVVH(D)F, continuous venovenous hemo(dia)filtration.

## Data Availability

Not applicable.

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
