# Peer review of "Aminoglycosides for the Treatment of Severe Infection Due to Resistant Gram-Negative Pathogens"

_antibiotics, 2023, doi:10.3390/antibiotics12050860_

Round 1

Reviewer 1 Report

Thank you for giving me the opportunity to review this article "Aminoglycosides for the treatment of severe infection due to resistant gram-negative pathogens". This article is very well comprehended. It would be appealing to readers if author can comment about newer agents in development like Plazomicin. Mechanism  of action and spectrum of activity was very written, could also include drug resistance and side effects of the aminoglycosides. Would suggest to include views about  combination therapy ( inhaled aminoglyosides with IV systemic antibiotics). 

Very well written with need for minor revisions 

Author Response

Q1.1. It would be appealing to readers if author can comment about newer agents in development like Plazomicin.

A1.1. We agree with the reviewer and thank him for his suggestion. We have added all these parts to the manuscript (lines 30 and 65).

Q1.2. Mechanism of action and spectrum of activity was very written, could also include drug resistance and side effects of the aminoglycosides.

A1.2. We have added a separate section for Spectrum and Side Effects. There is also a new table that summarizes AG side effects (lines 75 and 94).

Q1.3. Would suggest to include views about combination therapy (inhaled aminoglyosides with IV systemic antibiotics).

A1.3. Thank you for your comments. We have reviewed and explained the potential benefit of inhaled aminoglycosides as adjunctive therapy (line 341), but did not find significant data in combination therapy. If the reviewer has found more data, we will be happy to add it to the current review.

Reviewer 2 Report

This review article is interesting and worthy of publication after consideration of the following:

line 13: Gram-negative  bacteria infections

line 15  This article will review: delete “will” also  from L17 and L20

Enterobacterales & in vitro:  should be italicized through the manuscript

lines 55-58: please insert a reference

All abbreviations in Table 1(e.g. CVVH, RRT, BMI) should be written in the table footnote.

L175: Gram-negative 

L301: MDR GNB

All bacterial species in references should be italicized also the first letter of bacterial species shouldn’t be capital letter e.g.: L335: Pseudomonas aeruginosa & L343: Escherichia coli& L 534 and so on…..

Minor editing of the English language required

Author Response

Q2.1. line 13: Gram-negative bacteria infections / line 15  This article will review: delete “will” also  from L17 and L20 / Enterobacterales & in vitro:  should be italicized through the manuscript / lines 55-58: please insert a reference / All abbreviations in Table 1(e.g. CVVH, RRT, BMI) should be written in the table footnote. / L175: Gram-negative / L301: MDR GNB / All bacterial species in references should be italicized also the first letter of bacterial species shouldn’t be capital letter e.g.: L335: Pseudomonas aeruginosa & L343: Escherichia coli& L 534 and so on…

A2.1. We thank the reviewer for his careful proofreading of our manuscript. All the corrections were made.

Reviewer 3 Report

Dear authors,

This is a good paper about the current status of aminoglycosides in treating various infections. Although the paper is well structured, I have some recommendations to improve the manuscript:

1. Usually, the introduction ends with presenting the aims of the paper. Maybe you can briefly present this.

2. Table 1 -> Maybe you can consider adding some references to support your statements.

3. Lines 141,142 -> You say that "combining AG with β-lactams...enhance antibiotic resistance due to their synergistic antibacterial effects." 

In the article you cited [37], the authors observed synergistic effects and enhanced bactericidal activity, not "enhanced antibiotic resistance."

Please clarify this statement.

4. Maybe you can add some new references.

5. Line 3, 13, 18 -> Gram-negative with capital?

6. Line 256 -> Add a reference.

The paper requires only minor English editing.

Author Response

Q3.1. Usually, the introduction ends with presenting the aims of the paper. Maybe you can briefly present this.

A3.1. We have added the following sentence at the end of the introduction: “In this review, we will discuss the advantages and pitfalls of the most commonly used AGs in the treatment of severe resistant Gram-negative organisms, namely amikacin, gentamicin and tobramycin, as well as plazomicin, a promising next-generation aminoglycoside.”

Q3.2. Table 1 -> Maybe you can consider adding some references to support your statements.

A3.2. We thank the reviewer for his comment. We have added references in table 1.

Q3.3. Lines 141,142 -> You say that "combining AG with β-lactams...enhance antibiotic resistance due to their synergistic antibacterial effects." In the article you cited [37], the authors observed synergistic effects and enhanced bactericidal activity, not "enhanced antibiotic resistance." Please clarify this statement.

A3.3. We have clarified the statement and added a reference (line 214).

Q3.4. Maybe you can add some new references.

A3.4. We have added more recent references in the manuscript.

Q3.5. Line 3, 13, 18 -> Gram-negative with capital? / Line 256 -> Add a reference.

A3.5. All the corrections were made.